# Rational Design of a Skin- and Neuro-Attenuated Live Varicella Vaccine: A Review and Future Perspectives

**DOI:** 10.3390/v14050848

**Published:** 2022-04-20

**Authors:** Wei Wang, Dequan Pan, Tong Cheng, Hua Zhu

**Affiliations:** 1State Key Laboratory of Molecular Vaccinology and Molecular Diagnostics, National Institute of Diagnostics and Vaccine Development in Infectious Diseases, School of Life Sciences, School of Public Health, Xiamen University, Xiamen 361102, China; lukewang@xmu.edu.cn (W.W.); dequanpan@stu.xmu.edu.cn (D.P.); 2Department of Microbiology and Molecular Genetics, New Jersey Medical School, Rutgers University, Newark, NJ 070101, USA

**Keywords:** varicella-zoster virus, varicella, VZV, neuro-attenuated, ORF7, vaccine

## Abstract

Primary varicella-zoster virus (VZV) infection causes varicella, which remains a prominent public health concern in children. Current varicella vaccines adopt the live-attenuated Oka strain, vOka, which retains the ability to infect neurons, establish latency and reactivate, leading to vaccine-associated zoster in some vaccinees. Therefore, it is necessary to develop a safer next-generation varicella vaccine to help reduce vaccine hesitancy. This paper reviews the discovery and identification of the skin- and neuro-tropic factor, the open reading frame 7 (*ORF7*) of VZV, as well as the development of a skin- and neuro-attenuated live varicella vaccine comprising an *ORF7*-deficient mutant, v7D. This work could provide insights into the research of novel virus vaccines based on functional genomics and reverse genetics.

## 1. Introduction

Varicella zoster virus (VZV) is a neurotropic human α-herpesvirus that causes varicella (or chickenpox) during the primary infection, and then establishes lifelong latency in neurons of the dorsal root, cranial, enteric and autonomic ganglia [1]. Years later, the dormant VZV can reactivate to cause herpes zoster (HZ, or shingles) [2]. Varicella is a common and highly contagious skin disease in children, but its symptoms are usually mild and self-limiting. HZ is an acute infectious disease commonly seen in the elderly and in individuals with human immunodeficiency virus (HIV) infection, cancer, or those treated with corticosteroids and other immunosuppressive agents. The HZ incidence rate increases significantly with age. HZ is characterized by painful, unilateral clustered vesicular rashes, and is sometimes complicated by post-herpetic neuralgia (PHN), a refractory chronic pain syndrome persisting for months or longer after the HZ rash has resolved [2]. In addition, VZV infection may cause serious, life-threatening complications, and even death, in newborns, adults and immunocompromised or immunodeficient individuals [3].

## 2. Current Status of Varicella Vaccine

Vaccination is a cost-effective way to prevent varicella. Currently, only live-attenuated vaccines are available for protection against varicella, and the most widely used varicella vaccines employ the Oka vaccine strain of VZV (vOka). vOka was developed by Michiaki Takahashi in 1974 through serial passages of the wild-type parent Oka (pOka) strain of VZV in cell cultures [4,5]. This method resulted in a mixture of virus haplotypes, but the mechanism of vOka attenuation is still not well-characterized [6,7]. The vOka varicella vaccine has been introduced into routine childhood immunization programs in many countries [8]. Over the last few decades, studies have shown that vOka varicella vaccines are generally safe and effective in preventing varicella-related morbidity and mortality in healthy children worldwide [9,10,11,12]. Furthermore, varicella vaccination results in a decreased incidence of HZ in children, and provides protection for susceptible adults or immunosuppressed patients who are at high risk of fatal varicella [12,13].

However, there have been concerns raised about the use of vOka varicella vaccines following the analysis of adverse events (AEs) reported in post-market surveillance [14,15,16,17,18,19,20,21,22]. One notable AE is vOka-mediated HZ, which is associated with vOka’s ability to infect human neurons, establish latency and subsequently reactivate [23,24]. Although lower HZ incidences have been reported in vaccinated children than in unvaccinated ones by several studies [18,25,26], vOka is associated with a substantial proportion of post-vaccination HZ cases. For example, in three safety surveillance studies of vOka-Varivax from Merck, polymerase chain reaction (PCR) analysis identified vOka in 40.7% (48/118) [14], 35.2% (57/162) [15] and 47.1% (8/17) [16], respectively, of all clinical specimens collected from the reported post-vaccination HZ cases; the remaining specimens were positive for wild-type VZV, tested negative, or were considered inadequate for testing. Furthermore, there have been a few cases of vOka-mediated severe meningitis which occurred following the onset of HZ in both once- and twice-immunized children [22]. The vOka varicella vaccine, which is recommended for children aged 12 months to 12 years, has been on the market for only about 30 years, while HZ mainly occurs in adults over 50 years old. Therefore, it will take decades to confirm whether varicella vaccination causes a rise in the incidence of vOka-mediated HZ among the older population. More importantly, it is desirable to develop a safer varicella vaccine that will help to improve the acceptance of varicella vaccination, in order to better establish herd immunity against varicella.

To date, concerns over the safety of vOka have led to many studies that have evaluated the use of inactivated viruses or recombinant viral subunits as vaccines against VZV infection [27,28]. Vaccine candidates of both types were generally safe, immunogenic and effective for the prevention of varicella in healthy or immunocompromised adults, but extensive efforts are still required to further evaluate their use in healthy children. On the other hand, advances in reverse genetics of VZV have enabled the identification and manipulation of virulence factors [29,30,31], thus providing an opportunity for rational design of a novel live-attenuated varicella vaccine.

## 3. Identification of *ORF7* as a Skin- and Neuro-Tropic Factor of VZV

VZV induces pathological abnormalities, mainly in human skin and nerve tissues, during infection. Characterizing the crucial genes related to the skin- and neuro-tropisms of VZV will aid in the understanding of VZV pathogenesis, and it will also facilitate the development of improved vaccines. However, VZV is strictly human-specific, and has a highly cell-associated nature in cultured cells. These qualities make it challenging to study VZV gene functions and distinguish VZV from the other α-herpesviruses, which can infect a wide range of host species and generate large amounts of cell-free particles, leading to a much earlier and clearer understanding of their replication and pathogenesis compared to VZV. Recently, thanks to advances in molecular virology and reverse genetics, VZV gene function in specific cells and tissues has been characterized extensively by mutagenesis of the VZV genome, through the use of human tissues cultured ex vivo or implanted in severe combined immunodeficient (SCID) mice in vivo (Figure 1) [32,33,34].

We aimed to identify the key viral virulence and tissue-tropic factor(s) for the rational design of a next-generation, safer, live-attenuated VZV vaccine. For this purpose, we used bacterial artificial chromosome (BAC) technology to develop an efficient method for generating recombinant VZV mutants [31,35]. To better monitor the growth of mutant viruses in vitro and in vivo, a BAC clone of the wild-type VZV pOka strain, carrying both green fluorescence protein (GFP) and luciferase (luc) reporter genes, was constructed, and PCR methods were utilized to replace each open reading frame (ORF) in the BAC clone with a selectable marker (e.g., the *galK* gene), via homologous recombination in *E. coli* [31,32]. The resulting BACs were isolated and transfected into human melanoma cells. A total of 26 single-ORF deletion mutants expressing GFP and luc were reconstituted, and then screened in ex vivo human skin organ cultures (SOCs) [32]. In addition to previously reported skin-tropic genes such as *ORF10* [36], *ORF14* [37] and *ORF47* [38], we identified *ORF7* as a novel gene essential for VZV replication and spread in human skin [32]. Subsequently, to identify any potential neurotropic factors of VZV, we screened 18 ORF deletion mutants, that can grow normally in either epithelial cells or fibroblasts, in differentiated neuroblastoma (SH-SY5Y) cells and human embryonic stem cell (hESC)-derived neurons cultured in vitro [33]. Among them, only the *ORF7* deletion mutant showed a severe growth defect in these neuronal cell models, and its neuro-attenuated phenotype was further confirmed in human fetal dorsal root ganglia (DRGs), both ex vivo and in vivo [33]. Therefore, we identified *ORF7* as the first, and currently only, known full-length neurotropic gene of VZV. Altogether, our work identified *ORF7* as a skin- and neurotropic factor of VZV, and we found that while retaining replication competence in other cell lines, an *ORF7*-deficient mutant that is defective in spread in human skin and neurons would be a potential live-attenuated vaccine candidate against VZV infection.

## 4. Development of an *ORF7*-Deficient, Live-Attenuated Varicella Vaccine, v7D

To construct the *ORF7*-deficient VZV vaccine virus, we employed a stop-codon mutation strategy to minimize changes to the original genome sequence of the wild-type virus [39]. Firstly, the 11 bp region downstream of the ATG start codon of *ORF7* was replaced with a three-frame stop-codon cassette in the BAC clone of the VZV wild-type pOka strain, for the abrogation of *ORF7* expression. Then, the loxP-flanked BAC vector containing the GFP expression cassette was excised from the viral genome by co-transfecting the purified BAC clone with a Cre expression vector into human fibroblasts, thus reconstituting the *ORF7*-deficient candidate vaccine virus, v7D. Compared with the original wild-type strain of VZV-pOka, there was no difference in the genome sequence of v7D, except for the stop-codon mutation within *ORF7* and a single 34 bp loxP site in the noncoding region between *ORF60* and *ORF61*. Whole-genome sequencing of serially passaged v7D showed that its genome was stable for at least 25 passages in cell culture.

To thoroughly confirm the skin- and neuro-attenuation phenotype of v7D, a variety of cell and animal models currently available for evaluation of VZV infection and tissue tropism were used [39]. Similar to the reported *ORF7* deletion mutant [32,33], the lack of *ORF7* expression prevents the replication and cell-to-cell spread of v7D in human skin and neuronal cells, both in vitro and in SCID-hu mice in vivo. In contrast, wild-type VZV and vOka established a robust lytic infection in these cells. In addition, v7D failed to infect DRGs in the guinea pig and cotton rat models of VZV infection, and showed no intrathalamic neurovirulence or repeated-dose systemic toxicity in non-human primates. These preclinical toxicity evaluations show that v7D is highly skin- and neuro-attenuated, and should have a reduced risk of vaccine-associated complications, thus meeting the critical safety criteria for a next-generation live-attenuated varicella vaccine.

Despite the attenuation phenotype, v7D exhibited wild-type growth in MRC-5 human fetal lung fibroblasts, allowing for the production of v7D as a live vaccine [39]. Furthermore, similar to vOka, v7D retains full lymphotropism, which may help the replication and dissemination of vaccine viruses in vivo, and can functionally activate human dendritic cells (DCs) to process and present viral antigens to T cells in vitro [39]. These properties indicate that v7D has the ability to function as a live vaccine and stimulate human adaptive immunities against VZV infection.

VZV only infects humans, and thus VZV vaccine studies are limited to the use of common experimental animal models, which lack virus–host interactions under natural infection scenarios, to obtain immunogenicity data as indirect clues to VZV vaccine efficacy in humans. For preclinical immunogenicity evaluation of v7D, small animal models, including mice, rats, guinea pigs and rabbits, as well as non-human primate models, were used [39]. Three subcutaneous immunizations with the same doses of v7D and vOka induced similar and long-lasting VZV-specific antibody and T-cell responses, suggesting that the immunogenicity of v7D was non-inferior to that of vOka in small animals. v7D was also immunogenic in non-human primates. These preclinical toxicity and immunogenicity data have supported the initiation of a first-in-human phase I trial of the v7D-based, live-attenuated varicella vaccine in China (NO. ChiCTR1900022284).

## 5. Future Directions

Our preclinical data suggest that v7D is a promising, safer, live vaccine candidate against varicella. Nevertheless, many questions remain and should be addressed in future research.

### 5.1. Safety of the v7D Varicella Vaccine in Humans

Through future clinical observations, the v7D vaccine is expected to show several safety advantages compared to the vOka vaccine, including lower frequency of both varicella-like rashes and vaccine-associated HZ cases. However, fully confirming the long-term safety of v7D will require decades of observation for possible reversion to wild-type virulence, latency and reactivation of the vaccine virus, and revealing all AEs associated with v7D vaccination.

The possibility of reversion to a wild-type phenotype always raises concerns over the use of live-attenuated vaccines. However, like all herpesviruses, the VZV DNA genome is highly stable, and we have not observed the occurrence of wild-type reversion of v7D during serial cell passaging and long-term in vivo experiments. An early study showed that an *ORF61* mutant with stop-codon mutation reverted to wild-type virus in cell culture, but this virus retained the coding sequence of a RING finger domain of pORF61, which could have contributed to its wild-type reversion [40]. For v7D, the stop-codon mutation abrogated the expression of the entire *ORF7*, which could be beneficial to the stability of the virus genome and its attenuation phenotype. Nevertheless, it is necessary to continue to evaluate the risk of wild-type reversion of v7D in subsequent studies.

In addition, we do not know whether v7D would establish neuronal latency, although v7D cannot establish an initial lytic infection in human neurons and is much less likely to reactivate from possible latency, if at all. Therefore, it will be an important research question to determine whether v7D establishes latency and expresses VZV latency-associated RNA transcripts (VLT) in human neurons, based on the in vitro and in vivo models that are available to study VZV latency.

### 5.2. Immunogenicity and Efficacy of the v7D Varicella Vaccine in Humans

v7D has distinct properties of attenuation that may enable it to induce different patterns of immune response, compared to that of vOka, in humans. Therefore, with reference to an early study of vOka [41], the dynamics of the immunity induced by v7D of different doses should be investigated in clinical trials to determine an appropriate dose for its clinical use. Furthermore, since a routine two-dose varicella vaccination schedule for children that has been applied in several countries has been shown to provide better protection against varicella, further reducing breakthrough cases and the risk of outbreaks [42,43], it would be necessary to initiate further research on multiple dosing of the v7D vaccine to obtain a satisfactory protective efficacy. In addition, anti-VZV serum antibody is always used as a reliable indicator for serological confirmation of vaccine-induced VZV-specific immunity in humans, but there is still a lack of cell-mediated immunity (CMI) analysis in this field. Future clinical research is also necessary to investigate CMI to v7D vaccination and to reveal its role in the efficacy of varicella vaccine.

### 5.3. Mechanisms of v7D Attenuation and Vaccine-Induced Protective Immunity

v7D is a pure and genetically defined VZV strain that has a replication defect in both human skin and neuronal cells due to the abrogation of *ORF7* expression. *ORF7* encodes a virion tegument protein that localizes to the Golgi apparatus in the infected cells [44], the secondary envelopment site for VZV virus particles. Studies have shown that the deletion of *ORF7* does not prevent viral entry, viral genome replication, viral protein expression or retrograde transport of virus particles from axon terminals to somata, but substantially affects syncytia formation, secondary envelopment and subsequent cell-to-cell spreading of progeny viruses in differentiated skin and neuronal cells in vitro [39,44,45]. Therefore, pORF7 seems to work in conjunction with other viral or cellular proteins and acts as an essential part to regulate cell-to-cell fusion and secondary envelopment of VZV virus particles in infected human skin and neurons. Previously, we found that pORF7 interacts with pORF53 and they co-localize in the trans-Golgi network (TGN) in VZV-infected cells [46]. Most recently, the structure of pUL51 in complex with pUL7, the pORF7 and pORF53 homologues from herpes simplex virus-1 (HSV-1), has suggested a conserved role for pORF7 and its homologues in promoting membrane scission during cytoplasmic envelopment of nascent virions [47]. Further evidence is needed to validate and thoroughly characterize these mechanisms.

In addition, to overcome the limitation of VZV’s strict human specificity, future research on the mechanisms of v7D attenuation and protective immunity induced by v7D vaccine could be complemented by work in non-human primate models of simian varicella virus, which recapitulates the clinical features of VZV infection in humans [48].

## 6. Conclusions

Varicella remains an important public health concern worldwide. The vaccine virus vOka retains neurovirulence and may cause vaccine-associated HZ in some vaccinees. Thus, a safer neuro-attenuated vaccine is needed to improve vaccine acceptance and aid in establishing herd immunity against varicella, together with vOka vaccines. Through a genome-wide mutagenesis screen, we identified *ORF7* as a skin- and neuro-tropic factor of VZV, and we generated an *ORF7*-deficient, pOka-derived live virus vaccine candidate, v7D. Compared to vOka, v7D has a reduced risk of vaccine-associated AE in humans, and shows similar immunogenicity in animal models. Future research should focus on testing the safety and effectiveness of the v7D vaccine, determining its immune procedure and dosage through clinical trials, and understanding the mechanism of v7D attenuation and protective immune response induced by this vaccine virus. Together, this knowledge could help guide the future use of v7D as a safe viral vector for the prevention and possible treatment of infectious diseases.

## Figures and Tables

**Figure 1 viruses-14-00848-f001:**
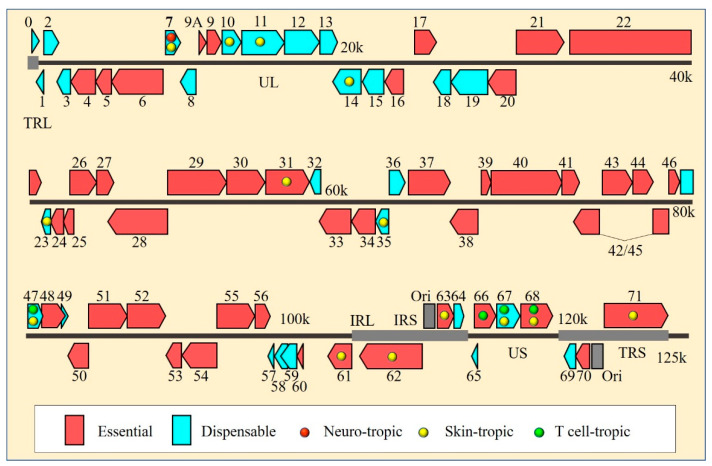
Genome-wide map of VZV genes required for virus replication and tissue tropism. Genomic arrangement of the open reading frames (ORFs) is based on the complete genome sequence of VZV pOka strain. VZV ORFs are colored according to their essentiality for virus growth in human melanoma cells, epithelial cells or fibroblasts in vitro. Essential and dispensable ORFs with functional domains that determine VZV tropisms are labeled with differently colored filled circles, depending on their specific function in human dorsal root ganglia (DRG), skin and T cells in vivo. The image is adapted with modifications from reference [32].

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
