# Peer review of "Rational Design of a Skin- and Neuro-Attenuated Live Varicella Vaccine: A Review and Future Perspectives"

_viruses, 2022, doi:10.3390/v14050848_

Round 1

Reviewer 1 Report

Thank you for asking me to review this paper on new insights in varicella immunization. The subject matter is certainly very important. Indeed, some unmet needs occur in this field: since the current vaccine virus vOka retains neurovirulence and may cause vaccine-associated HZ in some vaccinees, it is therefore not a suitable path towards the goal of eradication in the near future; moreover, live attenuated vaccines cannot be used in immunocompromised individuals. The Authors review the discovery and identification of the skin- and neuro-tropic factor ORF7 of VZV, alongside the development of a skin- and neuro-attenuated live varicella vaccine comprising an ORF7-deficient mutant, v7D.  The paper is clear and easy to read. The bibliography is sufficient and essential for what has been treated. However, the Authors limit their attention to the live attenuated vaccine only. It would instead be useful to the reader if the introduction briefly dealt with the potential of other types of vaccines, such as recombinant ones, given for example the evidence of immunocompromised seronegative immunization DOI: 10.1097/TP.0000000000003621

Reviewer 2 Report

The short article by Wang et al sent to Viruses summarizes the recent detailing of the use of an ORF7 expression defective virus. It is useful, although does not add much information beyond that included in the original paper that was published in Nature Communications in 2022.  The following comments are to improve the manuscript.

  1. It would be nice to add a figure of some sort. Maybe the gene position of ORF7 in regards to the layout of the other ORF7 genes
  2. There is nothing about the gene function of ORF7 that might be hinted at from work does on the corresponding gene of HSV, and indeed, HSV is not even mentioned, and given that HSV is so much better studied, is there anything that we can predict or model based on HSV studies?
  3. The authors promote the potential safety of the vaccine with a three frame stop codon in place rather than a full deletion. The logic of this strategy is counterintuitive to this reveiwer and should be explained.  a deletion could not revert.  however a simple three frame stop codon insertion could become WT if there is a single NT reversion.   There are similar strategies of doing the three frame stop codon strategy in which reversions have occurred; see Jeff Cohens ORF61 mutants, which reverted the stop codon in vivo cell culture.  This could very well occur with this virus, and it should thererfore be acknowledged as a potential safety issue, with the ORF61 reversion studies included, particularly in the discussion of safety section.
  4. The virus has a lox element in the 60-61 region. This could affect vlt expression. Although we know nothing of how VLT plays a role yet.
  5. Specific line comments
    1. 12/13, too many ‘ands’
    2. 14 vaccines should be vaccinees
    3. 25 there is evidence, (controversial as it may be) that VZV enters latency in autonomic and sympathetic ganglia and possible the CNS and enteric nervous system, and it sohlud be included
    4. 29 also those with cancer (Hogkins lypmoma) havce higher incidence of HZ
    5. 31 define what PHN is
    6. 39 “means is not the right word. I think “resulted in” is better
    7. 57 is this HZ cases or is this HZ like lesion cases, or is it varicella like vesicles ( I think it is the latter).
    8. 63/64 the sentence is not clear please rewrite.
    9. 88, 70 deletion mutants. – clarify as a considerable number of the BACs result in no virus growth,   and so mutants do not yet exist.
    10. 98 reference is given for the DRG model. It has nothing on ORF7, indeed, lines 100-102, A lot data is summarized but no reference ( I assume it is the nature communications paper)is given
    11. 162 I am not comfortable with the use of the word “unique” please rephrase  
